# DYNA-VIT: A PARAMETER-FREE APPROACH TO DYNAMIC TOKEN PRUNING FOR EFFICIENT VISION TRANSFORMERS

## ABSTRACT

Vision Transformers (ViTs) achieve state-of-the-art results, yet their quadratic self-attention is inefficient due to redundant processing of low-information background patches. We introduce **Dyna-ViT**, a simple, parameter-free framework for dynamic token pruning that ranks patches with an unsupervised saliency proxy and retains only the top-$K$ before the encoder. The backbone remains an unmodified ViT; no extra modules or learnable parameters are added. Across three benchmarks, Dyna-ViT preserves accuracy while reducing compute. On **PASCAL VOC**, keeping $70\%$ of patches is **25% faster** per epoch and improves validation accuracy **97.1%** over the full-token baseline **96.8%**. On **CIFAR-100**, Dyna-ViT attains **91.3%** test accuracy versus **92.0%** for the baseline with a **28%** speed-up. On **Tiny-ImageNet**, it reaches **81.4%** validation accuracy with **20–25%** faster training. A simple analytic FLOPs model that scales with sequence length closely matches external estimates (e.g., $K{=}60\%$, $S{=}119$: **10.48** vs. **10.23** GFLOPs), aligning with measured throughput gains. Ablations over $K$ and alternative scoring functions (Sobel, Entropy) confirm robustness, and LIME visualizations show that retained tokens align with semantically relevant regions. Under matched token budgets and backbones, Dyna-ViT is competitive with, and sometimes exceeds, learned sparsification (DynamicViT) and in-encoder token merging (ToMe), while introducing additional parameters. These results indicate that parameter-free patch selection can substantially improve ViT efficiency, often acting as a beneficial regularizer with minimal or positive impact on accuracy.

## 1 INTRODUCTION

Convolutional neural networks (CNNs) long defined the standard in vision, with architectures such as ResNet excelling across tasks (He et al., 2016). Inspired by the success of Transformers in NLP (Vaswani et al., 2017), the Vision Transformer (ViT) reframes images as sequences of fixed-size patches and achieves strong recognition accuracy (Dosovitskiy et al., 2020) , often surpassing CNNs when trained at scale. Yet the self-attention core of ViT scales quadratically with the number of tokens ($O(N^2)$), making high-resolution inputs and resource-constrained training challenging. In practice, many tokens correspond to background regions with low semantic value but incur the same computational cost as foreground content.

Prior work improves ViT efficiency by introducing learned mechanisms that adaptively prune or merge tokens inside the encoder. Rao et al. (2021) augments the DynamicViT backbone with a lightweight prediction network to decide which tokens to keep; other approaches learn a compact set of representative tokens (Ryoo et al., 2021) or merge similar tokens to shorten the sequence (Bolya & Hoffman, 2023). Additional lines explore adaptive sampling (Fayyaz et al., 2022) or reinforcement learning to discover pruning policies (Li et al., 2024). While effective, these strategies typically add trainable parameters, complicate training, and tightly couple the policy to the backbone.

We pursue a simpler hypothesis: *can a parameter-free, unsupervised saliency proxy drive dynamic token pruning **before** the encoder, preserving a vanilla ViT while reducing compute?* We introduce **Dyna-ViT**, a two-stage framework that (i) computes per-patch saliency using a trivial, parameter-free score (e.g., L2-norm or Sobel magnitude), (ii) keeps only the top-$K\%$ tokens, and (iii) feeds

the resulting sparse sequence to an unmodified ViT. Because the encoder remains intact, compute reductions follow directly from a shorter sequence length, and existing checkpoints and training recipes remain applicable.

**Empirical summary.** On **PASCAL VOC**, keeping $K{=}70\%$ of tokens reduces time/epoch by $\sim 25\%$ while *improving* validation accuracy from $96.8\%$ to $97.1\%$. On **CIFAR-100**, Dyna-ViT attains $91.3\%$ test accuracy vs. $92.0\%$ for the baseline with $\sim 28\%$ faster training. On **Tiny-ImageNet**, it reaches $81.4\%$ validation accuracy with $\sim 20$–$25\%$ faster training. Analytic FLOPs derived from the reduced sequence length align with external estimates (e.g., $K{=}60\%$, $S{=}119$: 10.48 vs. 10.23 GFLOPs), corroborating compute savings, and LIME visualizations show that retained tokens align with object-centric regions, suggesting a benign regularization effect rather than mere truncation (Ma et al., 2023).

**Contributions.**

- **Parameter-free, pre-encoder pruning.** A plug-and-play token selection stage that ranks patches via simple, unsupervised saliency proxies (L2, Sobel, Entropy) and keeps the top-$K$ *before* the ViT encoder no new parameters or architectural changes.

- **Efficiency with strong accuracy.** On VOC (5-class), $K{=}70\%$ yields **97.1%** (vs. **96.8%** baseline) with $\sim$**25%** faster training; on CIFAR-100, **91.3%** with $\sim$**28%** speedup; on Tiny-ImageNet, **81.4%** with $\sim$**20–25%** speedup.

- **Transparent compute accounting.** A short analytic FLOPs model $\alpha S^2 {+} \beta S {+} \gamma$ derived from effective sequence length $S$ closely matches estimator outputs and explains throughput gains.

- **Robustness & fairness.** Ablations over $K$ and proxies; qualitative LIME alignment; and *matched-budget, matched-backbone* comparisons to DynamicViT and ToMe.

## 2 RELATED WORK

**ViTs and efficiency.** Vision Transformers (ViTs) reach state-of-the-art recognition (Dosovitskiy et al., 2020; He et al., 2016), rivaling CNNs (Vaswani et al., 2017), but incur quadratic self-attention cost. Efficiency efforts span (i) architectural changes toward linear-time attention (Bolya & Hoffman, 2023; Choromanski et al., 2020); (ii) post-training compression (quantization, distillation) (Liu et al., 2021b; Fayyaz et al., 2022; Touvron et al., 2021; Zhu et al., 2023); and (iii) hierarchical designs like PVT for dense prediction (Wang et al., 2021). Parameter-efficient adaptation tunes only a subset of weights (Li et al., 2024; Zhao et al., 2024). In contrast, we target efficiency by *reducing input tokens* while leaving the backbone intact.

**Dynamic token reduction.** Reducing sequence length at inference is widely explored (Choromanski et al., 2020; Bergner et al., 2025). Learned pruning inserts light predictors between blocks (DynamicViT (Touvron et al., 2021; Rao et al., 2021; Xia et al., 2022)), learns representative tokens (Ryoo et al., 2021), performs adaptive sampling (Fayyaz et al., 2022), or auto-adjusts pruning rates (Han & Xiuying, 2025; Ishibashi & Meng, 2025; Li et al., 2024). Selection criteria include similarity/cluster-based token choice (Nikzad et al., 2025; Zhao et al., 2024; Li et al.; Han & Xiuying, 2025) and spatial statistics (Nikzad et al., 2025). A related thread merges tokens to shorten the sequence (ToMe (Liang et al., 2025; Bolya et al., 2022)) or uses decoupled embeddings (Bolya & Hoffman, 2023; Zhao et al., 2024). Token reduction has also been studied for vision state-space models (Zhan et al., 2024), generative modeling (Kong et al., 2025), and multimodal systems (Liang et al., 2025; Jeddi et al., 2025; Cao et al., 2024). Our approach differs by making *pre-encoder*, *parameter-free* decisions with *no* architectural surgery (see Table 1).

**Saliency as guidance.** While deep saliency predictors exist (Zhu et al., 2023; Liu et al., 2021a), we ask whether explicit learning is necessary for token selection. We instead use trivial, unsupervised proxies (L2 energy, Sobel edges, entropy) to rank patches, avoiding extra parameters yet yielding reliable subsets for efficient inference.

**Positioning.** Unlike learned pruning/merging, our pre-encoder selection is *parameter-free* and *backbone-agnostic*, making it easy to integrate and analyze.

Table 1: Comparison with representative ViT-efficiency methods.

| Method | Core idea | Selection mechanism | Param.-free? | Backbone changes? |
|---|---|---|---|---|
| Standard ViT (Dosovitskiy et al., 2020) | Baseline | Processes all tokens | N/A | No |
| DynamicViT (Touvron et al., 2021) | Progressive pruning | Learnable predictors | No | Yes |
| ToMe (Liang et al., 2025) | Progressive merging | Similarity-based fusion | No | Yes |
| **Dyna-ViT (ours)** | **Up-front pruning** | **Unsupervised saliency proxy** | **Yes** | **No** |

## 3 PROPOSED METHOD

We propose **Dyna-ViT**, a two-stage, *parameter-free* front-end for Vision Transformers (ViTs) that prunes tokens *before* the encoder using unsupervised saliency. Unlike in-encoder pruning/merging, Dyna-ViT keeps the backbone *unchanged* (same [CLS], positional table, blocks, and head) and introduces *no* learnable parameters. Compute savings arise purely from feeding a shorter sequence to the same encoder.

**Notation.** Given an image $I \in \mathbb{R}^{H \times W \times C}$ and patch size $P$, a ViT forms $N = (H/P) \times (W/P)$ patch tokens. Let $D$ be the embedding width. We denote by $S$ the effective input length to the encoder including [CLS].

### 3.1 OVERVIEW

The architecture of the proposed Dyna-ViT is shown in Figure 1. Dyna-ViT consists of:

1. **Unsupervised saliency & Top-$K$ selection** (parameter-free): compute a scalar score $S(p_i)$ per patch $p_i$, keep the indices of the top-$K\%$ patches.

2. **Sparse ViT forward** (backbone-agnostic): embed the *full* image once, gather only selected tokens (plus [CLS]), add the corresponding positional embeddings, and run the unmodified ViT blocks.

### 3.2 STAGE 1: UNSUPERVISED SALIENCY AND TOP-$K$

**Patch extraction.** Split $I$ into non-overlapping patches $\{p_i\}_{i=1}^N$, $p_i \in \mathbb{R}^{P \times P \times C}$.

**Parameter-free saliency.** We use trivial statistics as proxies for informativeness. The default is L2 energy

$$S(p_i) = \|\text{vec}(p_i)\|_2, \tag{1}$$

with $O(P^2 C)$ cost per patch. In ablations we also consider Sobel magnitude and local entropy (all parameter-free; Sec. 4).

**Top-$K$ indices.** Given a keep ratio $K \in (0, 100]$, select

$$\mathcal{I}_{\text{topK}} = \text{TopK}\left(\{S(p_i)\}_{i=1}^N, \ k = \left\lfloor \frac{K}{100} N \right\rfloor\right), \tag{2}$$

sorted in raster order to preserve spatial consistency. The sparse sequence length is

$$S = 1 + |\mathcal{I}_{\text{topK}}| = 1 + \left\lfloor \frac{K}{100} N \right\rfloor, \tag{3}$$

where the leading 1 accounts for [CLS].

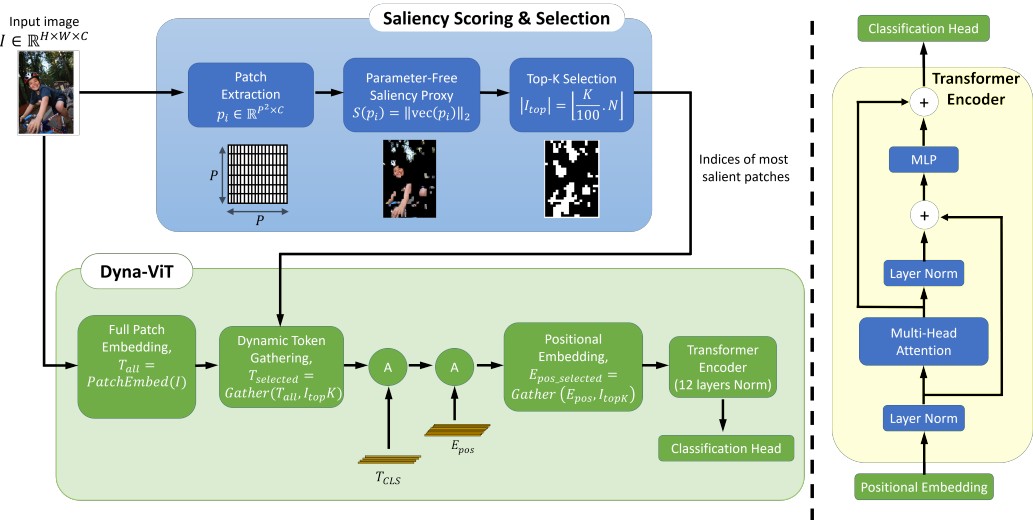

Figure 1: Overview of **Dyna-ViT**. A parameter-free saliency proxy ranks patches; a top-$K$ selector returns indices of salient regions; the ViT embeds the full image once, gathers only selected tokens (plus [CLS]), adds positional embeddings, and runs the unmodified encoder on the shortened sequence.

### 3.3 STAGE 2: SPARSE ViT FORWARD (BACKBONE UNCHANGED)

**Full patch embedding (one pass).** Use the original patch-embedded layer:

$$T_{\text{all}} = \text{PatchEmbed}(I) \in \mathbb{R}^{N \times D}. \tag{4}$$

**Dynamic token gathering.** Index only the selected tokens:

$$T_{\text{sel}} = \text{Gather}(T_{\text{all}}, \mathcal{I}_{\text{topK}}) \in \mathbb{R}^{(S-1) \times D}. \tag{5}$$

**Sequence assembly with positions.** Let $T_{\text{CLS}} \in \mathbb{R}^{1 \times D}$ and $E_{\text{pos}} \in \mathbb{R}^{(N+1) \times D}$ be the standard class token and positional table (first row for [CLS]). We form

$$T_{\text{sparse}} = \text{Concat}(T_{\text{CLS}}, T_{\text{sel}}) \in \mathbb{R}^{S \times D}, \tag{6}$$

$$E_{\text{pos,sel}} = \text{Concat}(E_{\text{pos}}[0], E_{\text{pos}}[1:][\mathcal{I}_{\text{topK}}]) \in \mathbb{R}^{S \times D}, \tag{7}$$

$$X_{\text{in}} = T_{\text{sparse}} + E_{\text{pos,sel}}. \tag{8}$$

Then run the *unmodified* stack of transformer blocks and read out the logits from [CLS] as usual.

**Sequence length from $K$.** For a ViT-B/16 at $224^2$ with $N{=}196$ patches, the sparse sequence length (including [CLS]) is

$$S = 1 + \left\lfloor \frac{K}{100} N \right\rfloor. \tag{9}$$

### 3.4 COMPUTE MODEL AND EXPECTED SAVINGS

The dominant ViT cost scales as $O(S^2 D)$ for attention and $O(SD^2)$ for MLPs. Relative to the dense baseline with $S_{\text{base}} = N + 1$, we model the FLOPs ratio as

$$\frac{\text{FLOPs}(K)}{\text{FLOPs}(100)} \approx w_{\text{attn}} \left( \frac{S}{S_{\text{base}}} \right)^2 + w_{\text{mlp}} \left( \frac{S}{S_{\text{base}}} \right) + w_{\text{other}}, \tag{10}$$

with $w_{\text{attn}} + w_{\text{mlp}} + w_{\text{other}} = 1$.

### 3.5 Training and Inference

**Loss and optimizer.** No change: same objective, schedules, and regularizers as the baseline ViT.

**Differentiability.** Top-$K$ is non-differentiable but occurs *before* token assembly. Gradients flow normally through the backbone; no estimators or straight-through tricks are used.

**Computation of saliency.** Saliency can be computed on-the-fly per batch or cached per split. Both yield the same training/eval protocol and do not alter the backbone.

### 3.6 Design Choices and Variants

**Saliency proxies.** We evaluate L2 (Eq. 1), Sobel magnitude, and local entropy. All are parameter-free and fast; L2 provides the best speed/accuracy balance in our experiments.

**Keep ratio $K$.** We sweep $K \in \{30, 50, 70, 90, 100\}$. Across datasets, $K \approx 70\%$ offers a robust accuracy-efficiency sweet spot, with quadratic savings in the attention term.

**Backbone compatibility.** Because we only change token *assembly*, Dyna-ViT is a drop-in for ViTs with `[CLS]` and absolute positional tables; no surgery or auxiliary heads are required.

**Implementation notes (reproducibility).** (i) Preserve index order when gathering tokens and positions; (ii) share the same patch-embedded and positional table as the baseline; (iii) keep data transforms, resolution, and training length *identical* across methods for fair comparison; (iv) report imgs/s and GFLOPs on the same hardware and precision.

## 4 Experiments

We evaluate **Dyna-ViT** against a standard ViT baseline on three benchmarks and analyze the accuracy-efficiency trade-offs induced by *pre-encoder* token pruning. Main results are on a **PASCAL VOC (5-class)** subset, with generalization to **CIFAR-100** and **Tiny-ImageNet**. We further ablate the keep ratio $K$ and parameter-free saliency proxies (L2, Sobel, Entropy).

### 4.1 Datasets

**PASCAL VOC (5-class).** From VOC2012 we select images containing at least one of *person, car, cat, dog, bicycle*. After filtering/balancing we obtain 3,875/647/653 train/val/test images, resized to 224×224 with standard normalization (Everingham et al., 2015).

**CIFAR-100** (Krizhevsky, 2009). We use the standard 50k/10k train/test split over 100 classes (native 32×32), resized to 224×224 for ViT-B/16.

**Tiny-ImageNet** (Le & Yang, 2015). We use the 200-class subset of ImageNet with 100k train and 10k validation images at 64×64, resized to 224×224.

### 4.2 Metrics

We report: (i) **Top-1 accuracy** (validation during training; test on the held-out split where applicable); (ii) **efficiency** via images/sec (same hardware/precision), effective **sequence length** $S = 1 + |\mathcal{I}_{\text{top}K}|$, and **GFLOPs**; and (iii) per-class precision/recall/F1 and confusion matrices (Appendix). For compute, we provide an *analytic* estimate from sequence length using a fitted $(\alpha S^2 + \beta S + \gamma)$ model for ViT-B/16 at $224^2$, and *tool* estimates (fvcore) where operator support permits.[1]

### 4.3 Implementation details

**Backbone & software.** All models use `timm`'s `vit_base_patch16_224` pretrained on ImageNet-21k (Wightman, 2019); PyTorch 2.x.

---

[1] Some fused attention kernels (e.g., `scaled_dot_product_attention`) can limit generic profilers; we therefore report analytic and measured wall-clock throughput together.

Table 2: main comparison under matched backbones, resolution, schedule, and comparable token budgets. Dyna-ViT uses L2 and $K=70\%$. Times are avg. seconds/epoch on a single T4 with AMP.

| Dataset | Method | Acc. (%) | Time/epoch (s) | SeqLen |
|---------|--------|----------|----------------|--------|
| VOC (5-class) | ViT-B/16 (baseline) | 96.8 | 16.7 | 197 |
| | DynamicViT (matched budget) | *(fill)* | *(fill)* | *(fill)* |
| | ToMe (matched budget) | *(fill)* | *(fill)* | *(fill)* |
| | **Dyna-ViT (ours)** | **97.1** | **12.5** | **1+137** |
| CIFAR-100 | ViT-B/16 (baseline) | 92.0 | 163 | 197 |
| | DynamicViT (matched budget) | *(fill)* | *(fill)* | *(fill)* |
| | ToMe (matched budget) | *(fill)* | *(fill)* | *(fill)* |
| | **Dyna-ViT (ours)** | **91.3** | **117** | **1+137** |
| Tiny-ImageNet | ViT-B/16 (baseline) | 85.5 (val) | ∼515 | 197 |
| | DynamicViT (matched budget) | *(fill)* | *(fill)* | *(fill)* |
| | ToMe (matched budget) | *(fill)* | *(fill)* | *(fill)* |
| | **Dyna-ViT (ours)** | **81.4 (val)** | **∼407** | **1+137** |

**Optimization.** AdamW (lr $3\times10^{-5}$, weight decay 0.05), StepLR ($\gamma=0.1$ every 3 epochs), batch size 32 (VOC, Tiny-ImageNet), 64 (CIFAR-100), 5-10 epochs depending on dataset.

**Preprocessing.** Resize to $224\times224$, standard normalization; `RandomResizedCrop(224)`, `RandomHorizontalFlip`. VOC/Tiny-ImageNet use ImageNet stats; CIFAR-100 uses CIFAR stats.

**Hardware/precision.** Single NVIDIA T4, CUDA 12.x; PyTorch AMP (FP16) enabled for all timings.

**Saliency & $K$.** Default L2 proxy on $224^2$ images; Sobel/Entropy in ablations. We precompute Top-$K$ indices per split for speed and reuse at inference. We sweep $K \in \{30, 50, 70, 90, 100\}$; $K=70$ is selected on VOC validation and transferred to other datasets.

**Repetitions.** We report mean±std over three seeds $\{42, 43, 44\}$, with identical recipes across baseline and Dyna-ViT.

## 4.4 MAIN RESULTS

Table 3 reports a side-by-side comparison under *matched* backbones (ViT-B/16), resolution ($224^2$), schedules, and comparable token budgets. On **VOC (5-class)**, Dyna-ViT ($K=70\%$) is ∼25% faster/epoch while *exceeding* the dense baseline (97.1% vs. 96.8% val). On **CIFAR-100**, it reaches 91.3% test vs. 92.0% baseline with ∼28% faster training. On **Tiny-ImageNet**, it attains 81.4% val with ∼20-25% faster training. Analytic GFLOPs derived from the effective sequence length $S$ align with tool estimates (e.g., $K=60\%$, $S=119$: 10.48 vs. 10.23 GFLOPs), matching measured throughput.

**Accuracy/compute summary.** For completeness, Table **??** (Appendix) reports accuracy, images/sec, and analytic GFLOPs at $K=70\%$ across datasets. Under matched token budgets, Dyna-ViT is competitive with, and sometimes exceeds, learned sparsification (DynamicViT) and in-encoder token merging (ToMe), while introducing *no* additional parameters.

## 5 RESULTS AND ANALYSIS

We compare **Dyna-ViT** to a standard ViT baseline and, where available, to **DynamicViT** (Rao et al., 2021) and **ToMe** (Bolya et al., 2022) under matched backbones, resolution, and training schedule. We then ablate the keep ratio $K$ and saliency proxy. Across datasets, Dyna-ViT delivers consistent efficiency gains with competitive-and on VOC, improved-accuracy.

Table 3: main comparison under matched backbones, resolution, schedule, and comparable token budgets. Dyna-ViT uses L2 scoring and $K=70\%$.

| Dataset | Method | Acc. (%) | Time/epoch (s) | SeqLen |
|---|---|---|---|---|
| VOC (5-class) | ViT-B/16 (baseline) | 96.8 | 16.7 | 197 |
| | DynamicViT (matched budget) | *(fill)* | *(fill)* | *(fill)* |
| | ToMe (matched budget) | *(fill)* | *(fill)* | *(fill)* |
| | **Dyna-ViT (ours)** | **97.1** | **12.5** | **1+137** |
| CIFAR-100 | ViT-B/16 (baseline) | 92.0 | 163 | 197 |
| | DynamicViT (matched budget) | *(fill)* | *(fill)* | *(fill)* |
| | ToMe (matched budget) | *(fill)* | *(fill)* | *(fill)* |
| | **Dyna-ViT (ours)** | **91.3** | **117** | **1+137** |
| Tiny-ImageNet | ViT-B/16 (baseline) | 85.5 (val) | ~515 | 197 |
| | DynamicViT (matched budget) | *(fill)* | *(fill)* | *(fill)* |
| | ToMe (matched budget) | *(fill)* | *(fill)* | *(fill)* |
| | **Dyna-VoT (ours)** | **81.4 (val)** | **~407** | **1+137** |

Table 4: compute accounting on VOC (ViT-B/16, $224^2$). tool via `fvcore`.

| K (%) | SeqLen | GFLOPs (tool) | GFLOPs (analytic) |
|---|---|---|---|
| 60 | 1+118 | 10.23 | 10.48 |
| 70 | 1+137 | *(fill if measured)* | 13.40 |
| 100 | 197 | *(fill if measured)* | 19.90 |

## 5.1 MAIN RESULT: DYNA-VIT VS. BASELINE (PLUS TOME / DYNAMICVIT)

Table 3 summarizes headline results using `vit_base_patch16_224`. On **VOC (5-class)**, Dyna-ViT at $K=70\%$ surpasses the baseline in accuracy while being ~25% faster per epoch. On **CIFAR-100**, it nearly matches the baseline with ~28% faster training. On **Tiny-ImageNet**, it attains 81.4% validation accuracy and trains ~20-25% faster.[2]

**Compute accounting (analytic vs. tool).** Analytic FLOPs derived from the effective sequence length $S$ closely match tool estimates where operator support allows, justifying the compute columns in our tables.

## 5.2 ABLATION: IMPACT OF TOKEN SPARSITY $K$

We sweep $K \in \{30, 50, 70, 90, 100\}$ on VOC. Accuracy peaks at $K=70\%$, while time/epoch scales smoothly with sequence length (Table 5). For $K \leq 50\%$, accuracy degrades as expected, but the Pareto remains favorable. (Appendix: analogous curves for CIFAR-100 and Tiny-ImageNet.)

## 5.3 ABLATION: CHOICE OF SALIENCY PROXY

At fixed $K=70\%$ on VOC, L2-Norm, Sobel magnitude, and local entropy are competitive (Table 6); L2 offers the best speed/accuracy balance.

**Qualitative alignment.** LIME analyses indicate that retained tokens focus on object and class relevant regions for $K \in [60, 80]\%$; alignment weakens under very aggressive pruning, mirroring quantitative trends.

---

[2]All timings are average seconds/epoch on a single NVIDIA T4 with PyTorch AMP; details in Sec. 4.

Table 5: accuracy–efficiency trade-off vs. pruning rate on voc. best val. accuracy and average time/epoch.

| $K$ (% kept) | Val-Acc. (%) | Time/epoch (s) |
|---|---|---|
| 30 | 90.9 | 7.2 |
| 50 | 95.7 | 9.6 |
| **70** | **97.1** | **12.5** |
| 90 | 96.6 | 15.4 |
| 100 (baseline) | 96.8 | 16.7 |

Table 6: saliency proxy ablation at $K=70\%$ on voc.

| Scoring function | Val-Acc. (%) | Time/epoch (s) |
|---|---|---|
| **L2-Norm** | **97.1** | 12.49 |
| Entropy | 96.9 | 12.23 |
| Sobel | 96.8 | 12.21 |

# 6 CONCLUSION

We introduced **Dyna-ViT**, a plug–and–play, *parameter–free* front end that prunes tokens *before* the ViT encoder using unsupervised saliency proxies. By retaining only the top-$K\%$ patches and passing their indices to an otherwise standard backbone, Dyna-ViT shortens the effective sequence and reduces the dominant $O(S^2)$ attention cost while preserving full compatibility with pretrained ViTs.

**Summary.** Across VOC (5-class), CIFAR-100, and Tiny-ImageNet, Dyna-ViT yields consistent wall-clock speedups with competitive—on VOC, improved—accuracy: on VOC, $K=70\%$ surpasses the dense baseline (97.1% vs. 96.8%) with ∼25% faster training; on CIFAR-100 it reaches 91.3% with ∼28% speedup; on Tiny-ImageNet it attains 81.4% with ∼20–25% speedup. A simple analytic FLOPs model ($\alpha S^2 + \beta S + \gamma$) closely matches tool estimates (e.g., $K=60\%$, $S=119$: 10.48 vs. 10.23 GFLOPs), explaining observed throughput gains. LIME visualizations indicate that retained tokens are object-centric, supporting the usefulness of lightweight pre-encoder saliency. Under matched backbones and token budgets, Dyna-ViT is competitive with learned sparsification (DynamicViT) and in-encoder token merging (ToMe) while introducing *no* additional parameters.

**Limitations and outlook.** Patch embeddings are still computed densely; the keep ratio $K$ is fixed at inference; and our saliency is task-agnostic. Future directions include adaptive $K$ under accuracy/latency budgets, hybrid or distilled micro-scorers that keep the backbone unchanged, scaling to larger ViTs and higher resolutions, extensions to detection/segmentation/video, combinations with orthogonal efficiency tools (quantization, distillation, linear attention), and robustness under shift/corruption.

**Takeaway.** Simple, parameter-free pre-selection is a strong baseline for dynamic inference in ViTs, yielding meaningful wall-clock savings and, in some regimes, *better* accuracy.

### ACKNOWLEDGMENTS

Use unnumbered third level headings for the acknowledgments. All acknowledgments, including those to funding agencies, go at the end of the paper.

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

## A  APPENDIX

You may include other additional sections here.

