# OpenReview forum: "Dyna-ViT: Parameter-Free Dynamic Token Pruning for Efficient Vision Transformers"
_ICLR.cc/2026/Conference — ICLR 2026 Conference Withdrawn Submission_

### Official Review · Reviewer_hkxa · 2025-10-26

**Soundness:** 3
**Presentation:** 3
**Contribution:** 1
**Rating:** 4
**Confidence:** 4

**Summary:**

This work proposes a parameter-free, pre-encoder token pruning method for VIsion transformers. Instead of modifying the ViT architecture or inserting learnable modules, the authors compute an unsupervised saliency score (e.g., L2 norm, Sobel edge magnitude, or entropy) for each patch, retain only the top-K% most salient ones, and feed this sparse token sequence into an unchanged ViT backbone.

**Strengths:**

The method is easy to understand, implement, and reproduce. The two-stage pipeline—saliency scoring + top-K selection—is transparent and avoids complex training or architectural changes. This method is also compitiable for most vision architectures with its pre-encoder prunning pipeline.

**Weaknesses:**

The idea (pruning low-saliency patches) has strong precedents in prior work such as EviT. The novelty and contributions are limited.

The keep ratio K is static per dataset, not adaptive per image. In heterogeneous datasets, this may over-prune simple images or under-prune complex ones.

Experiments are restricted to ViT-B/16 on relatively small datasets. It remains unclear how Dyna-ViT scales to larger models (ViT-L/H), higher resolutions (384+, 512+), or dense prediction tasks (detection, segmentation).

**Questions:**

The paper treats L2 norm, Sobel, and entropy as interchangeable heuristics—but doesn’t explain why these metrics correlate with semantic informativeness in ViTs.

With such a simple pipeline, can this method be used to more tasks and more senarios?

---

### Official Review · Reviewer_Kygw · 2025-10-27

**Soundness:** 1
**Presentation:** 1
**Contribution:** 1
**Rating:** 0
**Confidence:** 5

**Summary:**

This paper proposes Dyna-ViT, a parameter-free token pruning method for Vision Transformers. It selects the top-K% image patches using simple saliency measures (e.g., L2 norm or Sobel) and feeds them into an unmodified ViT to reduce computation. The authors report up to 25% faster training on small datasets, but comparisons with prior methods are all missing, and the writing quality needs significant improvement.

**Strengths:**

1.	The idea is very simple and easy to implement.

2.	The parameter-free setup can serve as a trivial baseline for related work.

**Weaknesses:**

1. __Incomplete experiments:__ The baselines are only two methods and all results of them are missing.

2. __Poor writing quality:__ Multiple placeholders and typos indicate an unfinished draft.

3. __Limited scope:__ it only experiments with small datasets without evaluations on ImageNet or real-world tasks.

4. __Weak novelty:__ the idea is a minor variation of existing dynamic ViT pruning techniques like EViT.

**Questions:**

The authors should complete the unfinished experiments, fill in all missing results, and polish the writing before considering submission.

---

### Official Review · Reviewer_YtMy · 2025-10-29

**Soundness:** 1
**Presentation:** 1
**Contribution:** 1
**Rating:** 0
**Confidence:** 5

**Summary:**

This work proposes a dynamic token pruning technique that ranks patches with an unsupervised saliency proxy and use that to prune tokens before the encoder. This is an incomplete work with many placeholders in the paper. The authors should have withdrawn the paper after the deadline and instead of wasting reviewers' time.

**Strengths:**

None

**Weaknesses:**

Incomplete submission.

**Questions:**

None.

---

### Official Review · Reviewer_5iuY · 2025-10-30

**Soundness:** 3
**Presentation:** 1
**Contribution:** 2
**Rating:** 0
**Confidence:** 5

**Summary:**

This paper proposes Dyna-ViT, a two-stage, parameter-free front-end for Vision Transformers (ViTs) that prunes tokens before the encoder using unsupervised saliency. It leverages parameter-free saliency measurements to select tokens prior to the Transformer backbone and aggregate unselected tokens into the remaining tokens.

**Strengths:**

* Very structured draft

**Weaknesses:**

* The most significant problem is that this paper is NOT completed. The experimental section and related tables read like a semi-finished article with lots of "fill" or "fill if measured". And many sentences are just fragments of words.

* In addition, this paper lacks experiments on standard-size dataset (such as ImageNet) and downstream tasks (such detection on MSCOCO), thereby lowering its significance.

* The idea of early exit and multi-branch processing are not novel. And the motivation of combining these two methods are weak.

* There is an obvious citation mistake in Table 1, where ToMe should be Bolya's work published in ICLR 2023.

**Questions:**

I suggest the authors to complete this manuscript first and then submit.

---

### Note · Authors · 2025-11-14

**Comment:**

I want to extend this work and cover all things according to suggestions and submit again.

**Withdrawal Confirmation:**

I have read and agree with the venue's withdrawal policy on behalf of myself and my co-authors.